# Correlation of the Handgrip Strength and Body Composition Parameters in Young Judokas

**DOI:** 10.3390/ijerph20032707

**Published:** 2023-02-02

**Authors:** Antonio Carlos Tavares Junior, Henrique Santos Silva, Tiago Penedo, Leandro George Spínola do Amaral Rocha, Alexsandro Santos da Silva, Rubens Venditti Junior, Júlio Wilson Dos-Santos

**Affiliations:** 1Postgraduate Program in Movement Sciences, São Paulo State University—UNESP, Bauru 17033-360, SP, Brazil; 2Laboratory and Research Group on Physiology Applied to Sports Training—FITES, Bauru 17033-360, SP, Brazil; 3Human Movement Research Laboratory—MOVI-LAB, Bauru 17033-360, SP, Brazil; 4Postgraduate Program in Human Development and Technologies, São Paulo State University—UNESP, Rio Claro 13506-900, SP, Brazil; 5Department of Physical Education, School of Sciences, São Paulo State University—UNESP, Bauru 17033-360, SP, Brazil

**Keywords:** martial arts, hand strength, dual X-ray absorptiometry, youth sports, athletic performance

## Abstract

Maximum isometric handgrip strength (MIHS) is a relevant parameter for judo performance and is related to health. Understanding the association between MIHS and MIHS relative (MIHSrel) and the absolute lean mass of the arm (LMarm) and the percentage of body fat (%BF) can provide important information for performance and health. The study aimed to investigate the correlation between MIHS and MIHSrel with the LMarm and the %BF of young judokas (sub-19, 15 males and 14 females). MIHS was measured using a multipurpose digital dynamometer with a load cell for computer. Body composition was measured by dual-energy X-ray absorptiometry (DXA). The correlation between MIHS, MIHSrel with LMarm and %BF was verified through Pearson’s correlation, with a significance level of *p* < 0.05. Correlation between MIHSrel and %BF was very high only in female judokas for both arms (right, r = −0.76; left, r = −0.75). Correlations between LMarm and MIHS of the right and left arms ranged from very high to almost perfect for both sexes (r = 0.74 to 0.94). These results highlight the importance of strengthening the arms in young judokas, and controlling body fat for performance and health, especially in female judokas.

## 1. Introduction

Judo is a combat sport characterized by an intense dispute for contact and control of the grip, corresponding to around 50% of the total fighting time [1,2,3]. Contact with the judogi (specific judo clothing) is essential to maintain control of the fight, attack, protect against striking, and promote the imbalance in the opponent. Besides that, the score will only be awarded when there is contact with the judogi [4,5]. Hence, handgrip strength has been identified as a relevant factor in judo performance, more precisely manifested in the conditions of resistance of isometric strength and maximum isometric strength [6]. Maximum isometric handgrip strength (MIHS) is positively correlated with body mass (BM) and forearm circumference [7,8], with BM being an important parameter for the evaluation of MIHS, mainly to estimate absolute values for comparative purposes [9]. Handgrip strength has also been pointed out as an important parameter for health [10,11]. Adults with lower handgrip strength are at greater risk of mortality [11] and associations between low levels of handgrip strength and premature death in young people have been found [12]. The frequent measurement of handgrip strength in sports and as a health parameter is based on several factors. First, handgrip strength is the most straightforward and least complicated of many instrumented muscle strength measures [10,11,13]. Second, there is some evidence that handgrip strength tends to reflect overall muscle strength [14,15]. Third, handgrip strength has clinical, comparative, and predictive value [10,11,14,15].

In this sense, judokas have higher MIHS than non-athletes [16], indicating that judo practice can promote handgrip strength and that this gain can remain stable over time. Indeed, a recent study suggests no difference in MIHS between judokas of different ages, i.e., Junior, Senior, and Master categories. The differences in this variable stem mainly from body mass [17]. Considering that fighting actions usually vary between genders and weight categories [3], strength parameters must be relativized, considering gender and weight categories. Relative maximum isometric handgrip strength (MIHSrel) has also been used to evaluate judokas using the MIHS/BM ratio [18,19]. Better performances of MIHSrel seem to be associated with a lower percentage of body fat (%BF) since judokas from lighter categories tend to present better MIHS results when relativized by BM [19]. This hypothesis is plausible because lighter judokas have a lower %BF [20], indicating that body composition can be an important indicator of MIHSrel. Since Judo is a sport divided into eight weight categories for Cadets (Under-18) and seven for Juniors (Under-21) and there are considerable anthropometric variations between these categories [20,21], BM and %BF are important variables in the analysis of MIHS and MIHSrel in youth judokas. Therefore, it is essential to know the characteristics of judokas in the different stages of preparation related to age groups, including cadets and juniors.

Although BM is a determining factor in strength, muscle mass determines the strength response. MIHS has also been used as a parameter to assess lean mass loss [15]. The forearm muscles are primarily responsible for handgrip [8,9], but the arm muscles are very important for performance, stances, and throws in judo [5,18,20]. In this sense, determining the absolute lean mass of the arm (LMarm) and %BF can bring important information to the understanding of the associations between these variables and MIHS and MIHSrel. A limitation of some studies on MIHS is that they need to use equipment that can measure specific body composition by body regions, as is done through dual X-ray densitometry (DXA). However, the use of DXA has not been employed in the relationship between strength and body composition in judokas. As a result, we did not find studies on the association between MIHS and MIHSrel and %BF and LMarm in judokas. Thus, considering the importance of parameters studied for health and performance and the gap in the literature on this topic, the present study aimed at investigating the correlation between the MIHS and MIHSrel with the LMarm and %BF and the influence of handgrip strength symmetry in young male and female judokas.

## 2. Materials and Methods

### 2.1. Sample

Twenty-nine judokas (15 male and 14 female) participated in this study. Males (17.3 ± 1.7 [CI95% 16.5–18.2] years, Stature = 173 ± 0.05 [CI95% 171–176] cm, Body Mass = 67.9 ± 10.8 [CI95% 62.5–73.4] kg) and females (17.7 ± 1.5 [CI95% 16.9–18.5] years, Stature = 160 ± 0.07 [CI95% 157–165] cm, Body Mass = 65.7 ± 11.03 [CI95% 60.1–71.6] kg). The participants were from state and national level in Brazil, training 5 times a week, approximately 3 h a day, with judo experience of 8.6 ± 0.8 years. The study was approved by the Ethics Committee of the School of Science at São Paulo State University (protocol code CAAE: 53686516.7.0000.5398), according to the declaration of Helsinki, and after the explanations of the study and the agreement of the coach responsible for the team were given. All participants were recommended not to drink alcoholic beverages, food supplements, stimulants, or any other type of substance 24 h before data collection, which was previously confirmed on each day of data collection. All judokas on the team agreed to participate in the study and before data collection they and their respective guardians, for those under 18, signed a consent form.

### 2.2. Procedures

The MIHS was measured using a multipurpose digital dynamometer, with a load cell for the computer, a precision of 0.1 kgf or 1 N, a total capacity of 200 kgf or 2000 N, and acquisition at a frequency of 50 Hz (CEFISE, Campinas, Brazil). The position adopted for the performance of MIHS followed the guidelines of the American Society of Hand Therapists [22]. The gripping handle was adjusted individually for each judoka according to each one’s footprint amplitude. Finally, the strength take-off was performed with the participant’s maximum possible force, in a sitting position, with the arm at the side of the body and the elbow positioned at 90°.

Three alternate measurements were taken with an interval of 10 s between them, for each hand. The largest measure among the three attempts was used. The ratio used to stipulate the MIHSrel was MIHS/BM.

Body composition was measured by dual-energy X-ray absorptiometry (DXA) with the Hologic Discovery Wi device (Hologic, Bedford, MA, USA). The reliability and validity of the equipment is high, specifically for judokas. The coefficient of variation has been reported to be 2.9% for fat mass and 1.7% for lean tissue and the validity of body composition is r > ~0.95, standard error of estimation (SEE) <1.98 [23]. The method estimated body composition by dividing the body into three anatomical compartments: fat-free mass, fat mass, and bone mineral content. The results were expressed in kilograms (kg) of absolute of the right and left LMarm, total body mass (BM), and % BF. Before each assessment, scan positioning, acquisition, and analysis were standardized following the manufacturer’s instructions.

### 2.3. Statistical Analysis

The normality of the data was verified using the Shapiro–Wilk test. After confirming the normality of the data, data were grouped and described in mean and standard deviation. The correlation between MIHS and MIHSrel with LMarm was verified through Pearson’s correlation. A paired t-test was performed to check for differences in LMarm, MIHS, and MIHSrel for the subjects’ right and left upper limbs. The effect size (ES) was calculated by Cohen’s d value and qualitatively interpreted using the following classification: 0.2, trivial; 0.2–0.6, small; 0.6–1.2, moderate; 1.2–2.0, large; 2.0–4.0, very large; and 4.0, extremely large [24]. The correlation was classified as (r): 0–0.1 very low; 0.1–0.3 low; 0.3–0.5 moderate; 0.5–0.7 high; 0.7–0.9 very high; 0.9–1.0 almost perfect [25]. Statistical analysis was performed using the statistical package SPSS Statistics software for Windows, version 22.0 (IBM Corp., Armonk, NY, USA). The significance level was set at *p* < 0.05.

## 3. Results

Anthropometric body composition and isometric handgrip strength are presented in Table 1. There was no difference between the right and left LMarm, both in males and females. However, in females, MIHS (*p* = 0.03; d = 0.63) and MIHSrel (*p* = 0.02; d = 0.70) presented higher values for the right arm. The correlations between the MIHSrel and the %BF showed significant very high negative correlations only in female judokas (Table 2). The correlations between LMarm and MIHS of the right and left arms ranged from very high to almost perfect for both genders (Table 3).

## 4. Discussion

The aim of this study was to investigate the correlation between MIHS and MIHSrel with the LMarm and %BF, considering the handgrip strength symmetry, in the results from young judokas. There was no difference between the right and left LMarm in males and females. Only the females presented differences in MIHS and MIHSrel with higher values for the right arm. The correlations between LMarm and MIHS of the right and left arms showed significant positive correlations for males and females (correlations very high to almost perfect), while between the MIHSrel and the %BF significant negative correlations were only in female judokas (correlations very high).

The %BF of male participants is compatible with that presented in other studies (12%) with judokas in the same age range [19,26]. However, the female’s group showed greater %BF than in the previous study [20,26], 28% vs. 23%, respectively. The correlation between %BF and MIHS were moderate in both men and women (no significance). These results corroborate the literature demonstrating that MIHS is dependent on the amount of BM and lean mass, regardless of the %BF [19,21]. However, in females, the correlations between %BF and MIHSrel had a very high negative significance for both arms. This data may be related to the larger %BF in the female group compared to males. Indeed, the relative strength is greater in the lighter categories of judokas, with the best rates recorded in the male categories ≤60 kg and ≤66 kg, decreasing as the weight class increases [19]. It is also known that judokas from the heavier categories have a higher %BF [20,26]. These two variables, %BF and MIHSrel, may be related to health risks, %BF and MIHSrel. A high %BF indicates obesity and being overweight, which increase the risk of cardiovascular diseases and cancer [27], and a lower MIHSrel is related to decreased strength, which is an important parameter for health in young people [12].

The body composition is a relevant factor for the performance of MIHSrel, and for other performance parameters since judokas of better level seem to have lower %BF [20,28] and heavier judokas have worse relative fitness scores than lighter judokas [28]. Therefore, we can assume that better athletic conditions are also related to better health conditions, considering the MIHSrel and %BF parameters. The MIHSrel is essential for performance in judo, as the sport is divided into weight categories. In addition, the grip is recognized as an element of paramount technical and tactical importance, being responsible for controlling the opponent and essential in attack and defense movements providing control and conditions to get a takedown [4,5,6], which explains the better handgrip strength of judokas compared to the general population [16].

Although there is no difference between right and left LMarm, female judokas showed differences in MIHS and MIHSrel with an advantage for the right arm. This strength difference may be related to unilateral training, prioritizing throws and stances to the right side. When evaluating the performance of young judokas in a specific test (Special Judo Fitness Test) performed with dominant and non-dominant sides, Simenko and Hadzic [29] found performance differences in favor of the dominant side (Total throws = 23.26 vs. 21.81, *p* < 0.003 and Index 15.34 vs. 16.42, *p* = 0.001). In addition, the authors found significant associations between the non-dominant side and the volume of competition and competitive results, showing the importance of using stances and throws for both sides, which increases the athlete’s effectiveness and enables greater technical-tactical variety, increasing the odds of winning. The capacity to attack from both sides, symmetrization, and a wider variety of techniques used in fights positively influence technical and tactical behaviors and the competitive results of youth judokas [30]. The symmetrical and healthy development of the musculoskeletal system and muscular strength are essential conditions for preparing youth judokas [29].

The judokas participating in this study have levels of handgrip strength compatible with what is expected for judokas in these age groups [17]. Concerning strength, the literature indicates a strong linear relationship between maximum MIHS and maximum upper and lower body strength in athletes of various sports [15]. For example, Schoffstall et al. [31] observed nearly perfect correlations (r = 0.97) between MIHS and powerlifting strength (i.e., bench press, squat, and deadlift). Furthermore, wrestlers and judokas with greater MIHS also showed greater overall strength (i.e., bench press, squat, and pull-up strength) and ballistic abilities (i.e., vertical jump, horizontal jump, sprinting, and shot-put performance) [32,33], supporting the hypothesis that MIHS is a predictor of overall strength and an important parameter for physical assessment. The importance of MIHS is also demonstrated by discriminating between wrestlers and judokas of elite and non-elite. For example, elite adult male wrestlers (ES = 1.17) and judokas (ES = 2.23–3.07) produced much larger MIHS than sub-elite adult male wrestlers and judokas [32,33]. Although some studies have failed to differentiate between medalists and non-medalists by the level of MIHS [34], there is a decrease in MIHS after successive competition fights, with high values of the rating of perceived exertion in the forearm by judokas and in the fingers by non-medalist judokas, which denotes a great demand for these structures and the importance of the handgrip [34]. Furthermore, the time spent during the fight in dispute for the grip is about 50% of the combat time and high MIHS rates were correlated to longer effective attack time, which is an important performance factor for a judo competition [35], which may be related to greater arms LMarm, when considering the results of the present study.

We found an almost perfect correlation for the right arm, a very high correlation for the left arm between LMarm and MIHS for the male group, and very high correlations for both arms in the female group for these same variables. Even though the forearm muscles are responsible for handgrip strength, our results suggest that the LMarm is an essential factor for the performance of MIHS. Simenko et al. [36] measured body symmetry/asymmetry in youth judokas and found differences in five out of fifteen body regions with a larger girth for the dominant (right) side, including forearm girth (*p* = 0.003) and elbow girth (*p* = 0.019), but not between arm girth. Together, our results and those from Simenko et al. [36] indicate less variation between the dominant and non-dominant sides of the LMarm and in the specific symmetrical development of these muscles and consequent MIHS as an adaptation to judo training.

Kubo et al. [37] compared three groups of judokas at different competitive levels verifying that judokas at the international level have a greater thickness of the extensor and flexor muscles of the elbow, normalized by height, when compared to judokas who had not yet reached international levels. However, there was no difference between groups, in muscle thickness, in the other seven points analyzed. Those results demonstrate the importance of LMarm for judo performance, as verified in our results by positive correlation between MIHS with LMarm. Thus, judo training causes adaptations that potentially increase MIHS and LMarm.

In addition to strength development being important for judokas, muscle strength is also important for adolescents in general. For example, a study that followed 1,142,599 adolescents, aged 16 to 19 years, for 24 years concluded that a low level of muscular strength in late adolescence, measured by knee extension and handgrip strength tests, is associated with all causes of premature mortality, especially cardiovascular disease, and suicide. Low levels of muscular strength can be compared to classic risk factors such as high body mass index or high blood pressure [12]. Another study with 165,000 athletes concluded that they live longer and have a reduced incidence of mortality from cardiovascular disease and cancer compared to the general population [38]. The high correlation between LMarm and MIHS found in our results (r = 0.74–0.94) corroborates the findings of studies that use MIHS as a parameter for detecting sarcopenia [10,11] and shows that judo is a sport which improves an important parameter related to health, such as handgrip strength.

Judokas have better MIHS rates than non-judokas [16]. Regarding the age group, after the category of cadets (Sub-18) there seems to be no difference between MIHS and the other categories (Junior, Senior, and Master), which demonstrates that the practice of judo develops MIHS in youth and can preserve MIHS until close to 60 years [17], indicating that judo contributes to a better physical condition and health of its practitioners from youth to old age.

In this sense, due to the increase in MIHS and its relationship with LMarm, judo contributes to a better physical condition and health of its practitioners from youth to old age. Thus, handgrip tests can also be used as a parameter to monitor the health of judokas beyond performance indicators.

In practice, measuring MIHS through dynamometry is an accessible, portable, reliable, and sensitive measure to detect performance metrics and make health-related prognoses. Although our results present practical information on training and health parameters for youth judokas, we understand the limitations of the present study, such as the correlation between LMarm with other variables of judo and analysis by weight category, since the sample did not have enough participants of all weight categories. Such limitations should be explored in future studies.

## 5. Conclusions

In conclusion, the present study found that the correlations between arms LMarm and MIHS varied between very high and almost perfect in both sexes. There is a very high negative correlation between MIHSrel and %BF in female judokas. Since handgrip strength is a parameter that assists technical, tactical, and physical fitness variables, arm exercises with the development of MIHS should have special attention in the preparation and development of judokas, as well as controlling body fat, especially in young female judokas. In addition, considering MIHS is an important health-related variable, judo practice may be a good option for health maintenance.

## Figures and Tables

**Table 1 ijerph-20-02707-t001:** Body composition and isometric handgrip strength.

	Male	Female	General
BMI (kg∙m^2^) [CI_95%_]	22.5 ± 2.9 (21.1–23.9)	25.4 ± 3.3 (23.6–27.2)	23.9 ± 3.4 (22.7–25.2)
Body Fat (%) [CI_95%_]	11.9 ± 2.6 (10.6–13.3)	27.9 ± 6.8 (24.4–31.5)	19.7 ± 9.6 (16.2–23.2)
LM_arm_ (kg)—Right [CI_95%_]	3.3 ± 0.5 (3.1–3.5)	2.2 ± 0.3 (2.1–2.4)	2.8 ± 0.7 (2.5–3.0)
LM_arm_ (kg)—Left [CI_95%_]	3.3 ± 0.5 (3.1–3.6)	2.3 ± 0.3 (2.1–2.4)	2.8 ± 0.7 (2.6–3.1)
MIHS (N)—Right [CI_95%_]	436.5 ± 63.8 (404.2–468.8)	366.9 ± 70.1 ^a^ (330.1–403.6)	402.9 ± 74.6 (375.7–430)
MIHS (N)—Left [CI_95%_]	418.8 ± 58.5 (389.2–448.5)	346.5 ± 64.6 (312.6–380.3)	383.9. ± 70.7 (358.2–409.6)
MIHS_rel_ (N∙kg^2^)—Right [CI_95%_]	6.46 ± 0.55 (6.18–6.74)	5.73 ± 1.47 ^a^ (4.95–6.5)	6.11 ± 1.1 (5.69–6.52)
MIHS_rel_ (N∙kg^2^)—Left [CI_95%_]	6.21 ± 0.59 (5.91–6.51)	5.39 ± 1.3 (4.71–6.07)	5.81 ± 1.1 (5.43–6.2)

BMI (Body Max Index), LM_arm_ (lean mass of the arm), MIHS (Maximal Isometric Handgrip Strength), MIHS_rel_ (relative isometric handgrip strength), CI_95%_ (95% confidence interval). Values are mean ± SD. ^a^ Significant difference between right and left arm by gender, *p* < 0.05.

**Table 2 ijerph-20-02707-t002:** Correlation between body fat percentage, maximum isometric, and relative handgrip strength of arm.

	Right Arm (r-Value)		Left Arm (r-Value)	
Male	MIHS	MIHS_rel_	MIHS	MIHS_rel_
%BF	0.39	−0.49	0.46	−0.32
Female	MIHS	MIHS_rel_	MIHS	MIHS_rel_
%BF	−0.45	−0.76 ^b^	−0.38	−0.75 ^b^

%BF (Body Fat), MIHS (Maximal Isometric Handgrip Strength), MIHS_rel_ (relative isometric handgrip strength). ^b^ Significant correlation, *p* < 0.01.

**Table 3 ijerph-20-02707-t003:** Correlation between lean mass and maximum isometric and relative handgrip strength of arms.

	Right Arm (r-Value)		Left Arm (r-Value)	
Male	MIHS	MIHS_rel_	MIHS	MIHS_rel_
LM_arm_	0.94 ^b^	−0.10	0.82 ^b^	−0.36
Female	MIHS	MIHS_rel_	MIHS	MIHS_rel_
LM_arm_	0.74 ^b^	0.36	0.75 ^b^	0.30

LM_arm_ (lean mass of arm), MIHS (Maximal Isometric Handgrip Strength), MIHS_rel_ (relative isometric handgrip strength). ^b^ Significant correlation, *p* < 0.01.

## Data Availability

Data from the present study can be obtained through the email professorjuniortavares@hotmail.com.

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
