# Peer review of "Correlation of the Handgrip Strength and Body Composition Parameters in Young Judokas"

_ijerph, 2023, doi:10.3390/ijerph20032707_

Round 1

Reviewer 1 Report

Review: IJERPH

Correlation of the Handgrip Strength and Body Composition Parameters in Young Judokas

General comments

Dear authors, I have some remarks that make me unable to endorse this study for publication at this stage. First, considering your aim, sample, variables, methods and results, discussing health-rated issues, in my opinion, is not appropriate in your study at all. Second, I completely miss rationale why this study needs to conducted from Introduction. Third, not controlling weight categories is big limitation which can affects results largely. Fourth, results are poorly presented. Fifth, discussion is very very weak and should be systematically rewritten. All this should be addressed and manuscript should be reviewed again.

Specific comments

Introduction

I miss brief literature overview from previous studies which investigated correlations between MIHS and MIHSrel with the LMarm and %BF. You stated “scarcity of studies”, but if there are no studies on this issue, and if this is the first one, I suggest to emphasize it. Also, I completely miss rationale for this study. It is very logical that arm lean body mass will be associated with MIHS. Why this should be investigated? In addition, I miss the explanation of the importance of researching young judokas.

Materials and Methods

In my opinion, sample should be better explained. What are the exclusion and inclusion criteria? What about weight categories, why you did not control it? What about age groups?

Procedures are well-described, but require references.

Line 112-116: Identification of differences in the LMarm, MIHS, and MIHSrel for the subjects' right and left upper limbs is not within your aim. Although results are presented, this is not discussed. Therefore, rationale for including it is not provided.

Results

Please consider exclusion of presenting differences, according to the previous comment. Or adjust aim and discussion of the study.

In general, I am of the opinion that results are poorly presented. Should be presented more detailed.

Discussion

Line 147-150: Redundant sentence for the first paragraph of discussion. Instead, I suggest to summarize the main results/conclusions of the study here.

Line 151-153: In my opinion, redundant for discussion.

Line 153-154: No correlation is also important information, and should be discussed and explained (not in relation to the females). Specifically, you should provide exact reasons for no associations between %BF and MIHS among male young judokas.

Line 155-160: As you did not control for weight categories, this seems irrelevant explanation. You should provide exact reasons why female young judokas with greater %BF had lower MIHS.

Line 162-163: Sarcopenia is muscle loss with aging. You investigated 17-18 y/o athletes. It is complete overconclusion that “their decreased handgrip strength can be associated with sarcopenia”.

Line 164-168: I am sorry but I really cannot follow this part.

Line 169-171: Contradictory sentence. You said in one sentence that “maximum isometric strength for the grip in judo remains unknown” and that “maximum isometric strength is important to the judo grip”.

Line 171-182: This text is completely irrelevant for discussion.

Line 183-187: Same as previous comment. Presenting importance of MHIS should part of Introduction.

Line 189-192: These findings should be discussed. You should provide explanation.  

Line 192-193: Please comment “Line 162-163”.

Line 194-200: I am sorry but I really don’t get it how “comparison of three groups of judokas at different competitive levels” demonstrate that “training time causes morphological adaptations that potentially increase MIHS and LMarm”. Also, I can get it how is this related to your results.

Line 200-204: Same here. Completely irrelevant and redundant to your study.

Line 205-212: This text is irrelevant for discussion. Presenting importance of MHIS should part of Introduction.

Line 213-224: Considering your aim, sample, variables, methods and results, discussing health-rated issues, in my opinion, is not appropriate in your study at all.

Line 225-229: Redundant for discussion.

Line 229-232: There are much more limitations. Also, you should provide rationale why you did not control judokas of different weight categories.

Conclusions

Should be rewritten after revision of discussion. 

Author Response

LETTER FOR EDITOR AND REVIEWERS

Dear Editor

We forward the responses to the Reviewers below. We heed all reviewer suggestions. We hope that our response to reviewers is still on time. Changes made were tracked using MS Word's “Track Changes” function.

Best Regards!

Reviewer

We would like to thank you for suggestions and comments on the manuscript. All suggestions were taken into account, and questions answered in response to your opinion. The considerations made the manuscript more robust and with better quality.  Thus, we corrected some points previously mentioned and we hope to have answered all the suggestions and clarified all the doubts, according to the answer to the opinion below.

Best Regards!

Comments:

General comments

Dear authors, I have some remarks that make me unable to endorse this study for publication at this stage. First, considering your aim, sample, variables, methods and results, discussing health-rated issues, in my opinion, is not appropriate in your study at all. Second, I completely miss rationale why this study needs to conducted from Introduction. Third, not controlling weight categories is big limitation which can affects results largely. Fourth, results are poorly presented. Fifth, discussion is very very weak and should be systematically rewritten. All this should be addressed and manuscript should be reviewed again.

Answer: Dear reviewer, we would like to clarify that we understand your point of view regarding the limitations of this study for the health area. Initially, this study was designed for the performance area. However, we were invited by the editor of the Journal to write a manuscript for this special issue. We think that this material, despite its limitations, has the potential to make a connection between performance and health that is something rare and that needs to be better discussed. We hope that after specific corrections, your opinion about the manuscript may change.

Specific comments

Introduction

I miss brief literature overview from previous studies which investigated correlations between MIHS and MIHSrel with the LMarm and %BF. You stated “scarcity of studies”, but if there are no studies on this issue, and if this is the first one, I suggest to emphasize it. Also, I completely miss rationale for this study. It is very logical that arm lean body mass will be associated with MIHS. Why this should be investigated? In addition, I miss the explanation of the importance of researching young judokas.

Answer: Our study is the first to verify the correlation between MIHS and MIHSrel with LMarm and %BF in judokas. This information was included in the Introduction before the objectives. It appears evident that forearm muscle mass correlates with MIHS/MIHSrel and not arm muscle mass, which seems to be more associated with elbow flexion and extension. Therefore, we explain better the importance of researching young judokas. Please, see lines 57-59; 67-69; 72-73 and 78-84. 

Materials and Methods

In my opinion, sample should be better explained. What are the exclusion and inclusion criteria? What about weight categories, why you did not control it? What about age groups?

Answer: We explain the sample indicating: training time, frequency, training volume and competitive level. Judokas are part of the same competitive judo club and have a similar level and experience. Judokas were not divided by weight categories because there would not be judokas in all categories. There was not a considerable variation in weight between the judokas (see 95%CI) nor for age (see 95%CI). All judokas belonged to the Cadets and Juniors categories. The information above is already in the text. However, evaluating the MIHSrel is a way to standardize the analysis considering the Body Mass of the participants.

Procedures are well-described but require references.

Answer: References were included. Please, see lines 106 and 117.

Line 112-116: Identification of differences in the LMarm, MIHS, and MIHSrel for the subjects' right and left upper limbs is not within your aim. Although results are presented, this is not discussed. Therefore, rationale for including it is not provided.

Answer: We added the aim of symmetry (line 84), and the results were discussed. See lines 259-269.

Results

Please consider exclusion of presenting differences, according to the previous comment. Or adjust aim and discussion of the study. In general, I am of the opinion that results are poorly presented. Should be presented more detailed.

Answer: As mentioned in the previous answer, those results were discussed.

Discussion

Line 147-150: Redundant sentence for the first paragraph of discussion. Instead, I suggest to summarize the main results/conclusions of the study here.

Answer: Thank you for your comments. Mains results/conclusions were included at the beginning of the discussion. Please, see line 172.

Line 151-153: In my opinion, redundant for discussion.

Answer: Information was deleted, and the content also was improved in the Discussion.

Line 153-154: No correlation is also important information, and should be discussed and explained (not in relation to the females). Specifically, you should provide exact reasons for no associations between %BF and MIHS among male young judokas.

Answer: Thank you for your comment. We have improved the discussion about the no associations between %BF and MIHS among male young judokas. Please, see lines 180-187.

Line 155-160: As you did not control for weight categories, this seems irrelevant explanation. You should provide exact reasons why female young judokas with greater %BF had lower MIHS.

Answer: From the description of the sample and average body mass (67.9 ± 10.8 [CI95% 62.5 – 73.4] kg) of the participants, it is possible to see that most are classified in the lighter categories.

Line 162-163: Sarcopenia is muscle loss with aging. You investigated 17-18 y/o athletes. It is complete over conclusion that “their decreased handgrip strength can be associated with sarcopenia”.

Answer: Thank you for your comment. That information was deleted. Please, see lines 192-197.

Line 164-168: I am sorry but I really cannot follow this part.

Answer: The discussion has been rewritten. Changes made were tracked using MS Word's “Track Changes” function. All your suggestions have been considered. Please, see lines 198-207.

Line 169-171: Contradictory sentence. You said in one sentence that “maximum isometric strength for the grip in judo remains unknown” and that “maximum isometric strength is important to the judo grip”.

Answer: The sentence has been deleted. See the rewrite of the text in the lines 214-228.

Line 171-182: This text is completely irrelevant for discussion.

Answer: The discussion has been rewritten. Changes made were tracked using MS Word's “Track Changes” function. All your suggestions have been considered. Please, see lines 241-249.

Line 183-187: Same as previous comment. Presenting importance of MHIS should part of Introduction.

Answer: Thank you for your comment. This was considered in the introduction.

Line 189-192: These findings should be discussed. You should provide explanation.  

Answer: These findings are part of previous studies and show how essential handgrip strength assessment is. Please, see lines: 229 – 249.

Line 192-193: Please comment “Line 162-163”.

Answer: The discussion has been rewritten. Changes made were tracked using MS Word's “Track Changes” function. All your suggestions have been considered. Please, see lines: 259-269.

Line 194-200: I am sorry but I really don’t get it how “comparison of three groups of judokas at different competitive levels” demonstrate that “training time causes morphological adaptations that potentially increase MIHS and LMarm”. Also, I can get it how is this related to your results.

Answer: That was an error. That information was corrected as per your suggestion. Please, see lines: 305 – 310.

Line 200-204: Same here. Completely irrelevant and redundant to your study.

Answer: Corrected as per your suggestion. Please, see lines: 281 – 293.

Line 205-212: This text is irrelevant for discussion. Presenting importance of MHIS should part of Introduction.

Answer: The initial part of the paragraph was deleted. A part was incorporated in the introduction, according to the suggestion. Please, see lines 57-59.

Line 213-224: Considering your aim, sample, variables, methods and results, discussing health-rated issues, in my opinion, is not appropriate in your study at all.

Answer: We previously addressed the perspective of the study proposal. We understand the problems in associating performance and health and try to respond to all the suggestions. The discussion has been rewritten. Changes made were tracked using MS Word's “Track Changes” function. All your suggestions have been considered. Please review.

Line 225-229: Redundant for discussion.

Answer: We deleted that text.

Line 229-232: There are much more limitations. Also, you should provide rationale why you did not control judokas of different weight categories.

Answer: More limitations were incorporated into the text with an explication about weight categories. We hope that after specific corrections, your opinion about the manuscript may change. Please, see lines 329-333.

Conclusions

Should be rewritten after revision of discussion. 

Answer: The conclusion has been rewritten. Other changes can be viewed using MS Word's “Track Changes” function.

Reviewer 2 Report

Dear Authors

You have written an interesting paper. However, some parts need to be addressed for greater clarity.

Introduction-

Lines 43-44 / what has sarcopenia and early diagnosis with youth judokas? This sentence is unnecessary and should be deleted. I haven't read any paper connected to youth judokas and sarcopenia - perhaps authors could point me to the literature.

Lines 59-62 - 7 weight categories are in seniors. Please refer to the number of weight categories for youth judokas (juniors and cadets)

Methods

Line 101 - please report the instructions one day prior to DXA measurements and the reliability/validity of the DXA machine.

Line 194 - correct referencing

Results - Table 1 / where is the paired T-test reported? It is unclear. Please amend

Paragraphs 189-193 / asymmetries from results should be better explained and reported. There are some papers that could help with the connection between morphological asymmetries (10.14589/ido.17.2.6). Also the perfect correlation between the left and right arm could be explained by the importance of bilateral throw execution. I recommend this paper to help back up your findings (https://doi.org/10.1123/ijspp.2021-0186).

Overall. The paper needs some English Proofreading and some additional work from the authors. However, the paper looks promising.

Kind regards

Author Response

LETTER FOR EDITOR AND REVIEWERS

Dear Editor

We forward the responses to the Reviewers below. We heed all reviewer suggestions. We hope that our response to reviewers is still on time. Changes made were tracked using MS Word's “Track Changes” function.

Best Regards!

Reviewer

We would like to thank you for suggestions and comments on the manuscript. All suggestions were taken into account, and questions answered in response to your opinion. The considerations made the manuscript more robust and with better quality.  Thus, we corrected all the points previously mentioned and we hope to have answered all the suggestions and clarified all the doubts, according to the answer to the opinion below.

Best Regards!

Comments:

Dear Authors

You have written an interesting paper. However, some parts need to be addressed for greater clarity.

Introduction

Lines 43-44 / what has sarcopenia and early diagnosis with youth judokas? This sentence is unnecessary and should be deleted. I haven't read any paper connected to youth judokas and sarcopenia - perhaps authors could point me to the literature.

Answer: The sentence has been altered.  Lines: 44-47.

Lines 59-62 - 7 weight categories are in seniors. Please refer to the number of weight categories for youth judokas (juniors and cadets).

Answer: Revised.

Methods

Line 101 - please report the instructions one day prior to DXA measurements and the reliability/validity of the DXA machine.

Answer: Reported. The instructions one day prior to DXA measurements (page lines 95-98); reliability/validity of the DXA machine (lines115-117).

Results

Table 1 / where is the paired T-test reported? It is unclear. Please amend

Answer: Reported in the caption – line 154. In addition, the information also is in Methods (line 128).

Discussion

Line 194 - correct referencing

Answer: Revised – Line 270.

Paragraphs 189-193 / asymmetries from results should be better explained and reported. There are some papers that could help with the connection between morphological asymmetries (10.14589/ido.17.2.6). Also, the perfect correlation between the left and right arm could be explained by the importance of bilateral throw execution. I recommend this paper to help back up your findings (https://doi.org/10.1123/ijspp.2021-0186).

Answer: We appreciate the references. They improved the discussion of our data. Changes can be viewed using MS Word's “Track Changes” function. Lines: 214-218; 259-269.

Overall. The paper needs some English Proofreading and some additional work from the authors. However, the paper looks promising.

Answer: English has also been revised.  Thanks for the valuable suggestions.

Kind regards

Round 2

Reviewer 2 Report

Dear Authors

Thank you for addressing all of my questions and suggestions. The paper's quality and clarity improved. The discussion has been well-updated and better connects your results to current research.

Therefore, I recommend acceptance in its current form.

Kind regards